# Isolation and Identification of *Lactococcus lactis* and *Weissella cibaria* Strains from Fermented Beetroot and an Investigation of Their Properties as Potential Starter Cultures and Probiotics

**DOI:** 10.3390/foods11152257

**Published:** 2022-07-28

**Authors:** Ewelina Maślak, Michał Złoch, Adrian Arendowski, Mateusz Sugajski, Izabela Janczura, Joanna Rudnicka, Justyna Walczak-Skierska, Magdalena Buszewska-Forajta, Katarzyna Rafińska, Paweł Pomastowski, Dorota Białczak, Bogusław Buszewski

**Affiliations:** 1Department of Environmental Chemistry and Bioanalytics, Faculty of Chemistry, Nicolaus Copernicus University in Toruń, Gagarina 7 Str., 87-100 Toruń, Poland; e_maslak@doktorant.umk.pl (E.M.); jrud@umk.pl (J.R.); walczak-skierska@umk.pl (J.W.-S.); katraf@umk.pl (K.R.); bbusz@chem.umk.pl (B.B.); 2Centre for Modern Interdisciplinary Technologies, Nicolaus Copernicus University in Toruń, Wileńska 4 Str., 87-100 Toruń, Poland; adrian@arendowski.hub.pl (A.A.); mateusz.sugajski@o2.pl (M.S.); p.pomastowski@umk.pl (P.P.); 3Polmlek Grudziądz Sp. z o.o., Magazynowa 8 Str., 86-302 Grudziądz, Poland; i.janczura@vp.pl (I.J.); d.bialczak@polmlek.com (D.B.); 4Institute of Veterinary Medicine, Faculty of Biological and Veterinary Sciences, Nicolaus Copernicus University in Toruń, Lwowska 1 Str., 87-100 Toruń, Poland; m.buszewska@umk.pl

**Keywords:** lactic acid bacteria, laser desorption/ionization, B vitamins, metabolites

## Abstract

The presence of certain microorganisms in dairy products or silage is highly desirable. Among them are probiotic strains of lactic acid bacteria (LAB), which show many beneficial features, including antimicrobial properties that support the development of beneficial microflora; in addition, owing to their biochemical activity, they influence the nutritional, dietary, and organoleptic properties of food products. Before being placed on the market, each strain requires separate testing to determine its probiotic properties and effectiveness. The aim of this study was to isolate LAB strains from a pickled beetroot sample that could be used in the dairy industry and with the potential to be considered as a probiotic in the future. Two strains identified using the MALDI technique were selected—*Lactococcus lactis* and *Weissella cibaria*. The optimal growth conditions of the strains were determined, and their proteolytic properties were assessed with the use of the o-PA reagent and spectrophotometry. The lipid profile was analyzed using the SALDI (surface-assisted laser desorption/ionization) technique and silver nanoparticles. High-performance liquid chromatography was used to assess the ability of the strains to synthesize beneficial metabolites, such as B vitamins (B2, B3, and B9) or lactic acid, and gas chromatography was used to analyze the substances responsible for organoleptic properties. Moreover, the ability to inhibit the growth of pathogenic strains was also tested in the selected strains. Both tested strains demonstrated the desired properties of starter cultures for future use in functional food production, showing that fermented plant products can serve as valuable potential probiotic sources.

## 1. Introduction

Probiotics have been around humankind since people started consuming fermented milk and food. Nevertheless, their beneficial effects remained unexplored until the last century. During this time, the meaning of the word has been changed several times; however, it has always been equated with health benefits. Currently the FAO/WHO recommends using the word probiotic in the context of “live microorganisms which when administered in adequate amounts confer a health benefit on the host” [1]. The effectiveness of the action of probiotic microorganisms depends on the size of their populations, which the definition does not specify. It is assumed that probiotic food products should contain more than 10^6^–10^7^ colony forming units (CFU) per gram so that the consumer can easily supply 10^9^ CFU per day [2,3]. During food production, in order to accelerate or start the fermentation process, starter cultures are added to raw products in addition to probiotics. Due to the fact that they exhibit enzymatic activity, both probiotics and starter cultures enable the desired properties to be obtained in the final product (e.g., flavor and texture) [4].

Many microorganism species can be used as a probiotic or starter culture, including bacteria, yeasts, and molds. These microorganisms are added or can be found naturally in many different foods, such as fermented dairy products (yoghurt, cheese, and kefir), pickled vegetables and fruits, cereal, fermented sausages, bread, beer, or wine [2,5]. In the food industry, lactic acid bacteria (LAB) predominate. They mainly include the following genera: *Lactobacillus*, *Lactococcus*, *Enterococcus*, *Streptococcus*, *Pediococcus, Leuconostoc*, and *Weisella* [6]. A common feature of all LAB is the ability to anaerobically metabolize a number of sugars via fermentation (depending on the species/strains). The main product of this process is lactic acid, which has preservative properties and contributes to the inhibition of the growth of pathogenic microorganisms. Homofermentative bacteria, such as those belonging to the *Lactoccocus* or *Streptococcus* genera, produce this acid via the Embden–Meyerhof (EM) pathway [7]. In contrast, heterofermentative LAB strains (*Weissella*, *Leuconostoc*, and certain *Lactobacillus*), in addition to lactic acid, can also produce acetic acid, ethanol, carbon dioxide, or acetate through the phosphoketolase pathways. Depending on inter alia temperature and the strain used, the amount of lactic acid produced may vary from a few to over 200 g per liter of medium [8]. Lactic acid neutralizes the electrochemical potential of cell membranes and denaturates the intracellular proteins of microorganisms by lowering the pH of the environment, which limits the growth of pathogenic and spoilage bacteria. In addition, the beneficial changes that take place in the host’s body under the influence of probiotics include an increase in the number of positive intestinal microflora. The antimicrobial activity of probiotic strains also results from their ability to form biofilms on the gut wall, compete for nutrients, and produce or release various metabolites (lactic acid, antimicrobial peptides, short-chain fatty acids, and hydrogen peroxide) [2,6]. The important properties of probiotics also include their ability to biosynthesize water-soluble B-group vitamins (folic acid, riboflavin, niacin, and cobalamin), which are not produced by the human body [7]. Each B-group vitamin is chemically different and acts in synergy to maintain the body’s homeostasis by playing major roles in metabolic processes such as energy production and red blood cell formation. These vitamins are easily removed or destroyed during cooking and food processing; thus, the use of bacteria makes it possible to enrich foods with these compounds in situ, which is an additional benefit for consumers. Moreover, LAB strains produce enzymes that cause the degradation of fats, proteins, and carbohydrates present in food products. As a result, volatile compounds, such as aldehydes, ketones, esters, or alcohols, are formed, which give the final food products specific organoleptic properties [9]. LABs are also a rich source of lipid compounds that form cell membranes, participate in metabolic pathways (in protein transport, DNA replication, etc.), and are a reserve energy material. Hence, it is important to analyze the qualitative and quantitative composition of bacterial lipid compounds in order to determine and characterize the unique properties of the LAB [10].

Many studies show that consuming products that contain probiotics contributes to the improvement of intestinal health by regulating the microflora and stimulating and developing the immune system. Their consumption has a beneficial health effect in the case of food and urogenital infections, colon cancerogenesis, coronary heart disease, blood pressure, diabetes and obesity, allergies [11,12], psychiatric disorders [13], and even vaccine responses [14]. The presence of L(+) lactic acid in probiotic products also contributes to the reduced absorption of toxic substances and cholesterol into the blood [15]. Increased consumer awareness of the positive properties of products containing probiotics has led to an increase in their consumption. This is related to the great interest of food producers (especially in dairy sectors) in searching for new strains or isolates that show the best probiotic properties. 

The aim of this study was to isolate LAB strains from a pickled beetroot sample that could be used in the dairy industry as starter cultures for future use in the production of functional foods. For the selected and identified LAB strains, tests were carried out to assess their physiological characteristics, interactions with pathogenic bacteria, ability to synthesize B vitamins and lactic acid, and production of volatile compounds.

## 2. Materials and Methods

### 2.1. Isolation of Microorganisms

The LAB strains were isolated from homemade pickled beetroots. Pickled foods are widely regarded in Poland as having health-promoting properties and a being a good source of natural probiotics. A series of 10-fold dilutions ranging from 10^−1^ to 10^−4^ in peptone water (Peptone Water, Sigma-Aldrich, Germany) was prepared from the tested samples. The samples and all dilutions were then applied to 7 different culture media. A total of 100 μL of the appropriate dilution was applied to the medium and evenly distributed with a spreader. The prepared plates were incubated for 24–72 h in three conditions: aerobic conditions at 37 °C, aerobic conditions at 25 °C, and anaerobic conditions at 30 °C. Then, based on morphological differences, single colonies of bacteria were selected, from which reduction cultures were prepared on the same media to obtain pure cultures. The following culture media were used: M-17 Agar (Sigma Aldrich, Steinheim, Germany), MRS Agar (Sigma Aldrich, Steinheim, Germany), China Blue Lactose Agar (Sigma Aldrich, Steinheim, Germany), APT Agar (Merc, Darmstadt, Germany), and TOS (Merck, Darmstadt, Germany).

### 2.2. Identification Using the MALDI Technique

The identification of the isolated strains was carried out using the MALDI (matrix-assisted laser desorption/ionization) spectrometric technique (microflex LT MALDI-TOF MS) by the direct application of the sample to the plate (according to the procedure recommended by the manufacturer [16]). For this purpose, the isolated colony was spread as a thin layer directly onto the sample position. Then, 1 μL of 70% formic acid was applied to the sample; after it dried, 1 μL of HCCA matrix solution (10 mg/mL of a solution containing 50% acetonitrile, 47.5% water, and 2.5% trifluoroacetic acid) was applied. The collected spectra were analyzed with the help of the Biotyper platform. From the isolated strains, 2 with the expected properties were selected for further analysis.

### 2.3. Microscopic Observation

For the selected strains, Gram staining was performed followed by a microscopic examination of the cells’ morphology according to the following procedure: (1) suspend bacterial cells in a drop of distilled water applied to a microscopic slide using a microbial loop (1 μL); (2) fix the bacterial cells after water evaporation by transferring the slide over a burner (3 times); (3) apply the basic dye (crystal violet; 3 min); (4) rinse with distilled water; (5) apply the iodine solution (2 min); (6) rinse with distilled water; (7) apply the ethanol (30 s); (8) rinse with distilled water; (9) stain with an additive colorant (safranin; 30 s); (10) rinse with distilled water. 

### 2.4. Confirmation of Identification by 16S rRNA Gene Sequencing

For DNA extraction, single colonies of the 2 selected strains were subcultured in liquid media (Tryptic Soy Agar, Sigma Aldrich, Steinheim, Germany) and incubated for 24 h at 30 °C. A total of 2 mL of liquid culture of each bacteria strain was centrifuged, and DNA was extracted using a combination of bead-beating and an E.Z.N.A. Bacterial DNA Kit (Omega BIO-TEK, Norcross, USA). The universal 16S rRNA bacterial primers 1492R (5′-GGTTACCTTGTTACGACTT-3′) and 27F (5′-AGAGTTTGATCCTGGCTCAG-3′) were applied to amplify this gene [17]. The PCR products were visualized on a 1% agarose gel stained with ethidium bromide under UV light to confirm the presence of an approximately 1350 bp band. The PCR products were purified and sequenced by Genomed (Warsaw, Poland). The obtained results were analyzed using the BioEdit program and compared with the NCBI online database.

### 2.5. Growth Assessment of Selected Isolates

The aim of this step was to find the best culture conditions for the cultivation of the selected bacteria in view of biomass preparation for future starter cultures. To determine the optimal growth conditions, bacterial suspensions with optical densities of 0.5 McF were prepared in liquid culture media. The following media were used: MRS broth (Sigma Aldrich, Steinheim, Germany), tryptic soy broth (Sigma Aldrich, Steinheim, Germany), M-17 broth (Sigma Aldrich, Steinheim, Germany), and LAPTg. The LAPTg medium was prepared by dissolving the following ingredients in 1 L of water: 10 g of yeast extract, 15 g of peptone, 1 mL of Tween 80, 10 g of tryptone, and 18 g of lactose (pH approx. 6.5). The bacterial suspensions were incubated at 20 and 30 °C. The optical density of each sample after 6, 24, and 48 h was measured in triplicate. A DEN-1 densitometer (Biosan) was used for the measurements. 

### 2.6. Determination of Proteolytic Properties

The proteolytic activity of the selected two strains was determined by means of the spectrophotometric test using the o-phthaldialdehyde (o-PA) reagent [18]. The o-PA solution was prepared by mixing the following reagents: 25 mL of 100 mM sodium tetraborate, 2.5 mL of 20% (m/V) sodium dodecyl sulfate, 1 mL of 40 mg/mL o-PA dissolved in methanol, and 0.1 mL of β-mercaptoethanol.

Before starting the tests, the bacteria were plated on solid MRS agar and incubated for about 24 h at 30 °C. The appropriate amount of biomass was then transferred to 0.9% saline via a loop to determine the optical density of 1 McF. Subsequently, 1 mL of the obtained suspension was transferred into 9 mL of 10% (m/V) RSM (reconstituted skimmed milk) and incubated at 37 °C for 24 h. After the incubation, 0.75 M trichloroacetic acid was added to the obtained cultures for deproteinization at a volume ratio of 1:2. The samples were centrifuged, and the obtained supernatants were incubated in o-Pa solution for 10 min at room temperature. Then, the absorbance (λ = 340 nm) was measured against the control sample using a UV-Vis spectrophotometer (NanoDrop 2000c, Thermo Scientific). Each sample was measured in triplicate. The results were expressed in millimoles (mM) of free amino acids (FAA) per liter of milk by referring to a prepared standard curve of glycine.

### 2.7. Volatile Organic Compounds Analysis

MRS agar was poured into sterile headspace tubes. The selected bacterial strains were plated on the medium and incubated at 37 °C for 48 h. Subsequently, solid phase microextraction (SPME) was performed using Carboxen/polydimethylsiloxane fiber. The extraction was performed for 15 min, and desorption was carried out in the dispenser for 2 min at 230 °C. A chromatographic analysis was performed using GC-MS (Agilent) with a splitless dispenser and a temperature gradient program. The temperature program for the chromatographic furnace was as follows: 40 °C (2 min), accretion at 10 °C/min to 140 °C (10 min), and accretion at 5 °C/min to 270 °C (5 min). The flow rate of the helium carrier gas through the column was 1.2 mL/min. Electron beam ionization (EI) at 70 eV was used. The temperature of the ion source was 250 °C and the temperature of the transfer line was 300 °C. MS spectra were recorded in the range of 35–550 m/z. The obtained data were deconvolved and equalized. Integrated peaks were identified using the NIST 17 library. The following quality control criteria were used: matching factor min. 70% and presence in min. 2 out of 3 analyzed repetitions.

### 2.8. Analysis of Fats by Mass Spectrometry

The two selected strains were analyzed and cultured for 48 h at 37 °C in liquid medium (MRS broth). After centrifugation, the biomass was washed in 1 mL of 0.9% NaCl solution, then lipid extraction was performed using the Folch method. A total of 1.5 mL of a 2:1 (vol/vol) chloroform/methanol mixture was added to the biomass. The Eppendorf tubes were transferred to an ultrasonic bath and left for 15 min at room temperature. Then, 0.5 mL of 0.05 M NaCl was added to the suspension and vortexed for 10 min. In the next step, the suspensions were centrifuged for 15 min at 2415× *g*, the upper layer was collected in separate test tubes, 0.5 mL of 0.05 M NaCl solution was added, and the tubes were vortexed again. The two bottom layers were combined, and the solvent was evaporated using a vacuum centrifuge. The lipid samples were dissolved in chloroform and a volume of 0.5 µL was applied to the plates immediately before measurement.

A plate covered with a layer of silver nanostructures applied to the surface of H17 steel by means of electrodeposition was selected for the measurements [19]. Prior to the synthesis of silver nanostructures, the plates were cleaned in an ultrasonic cleaner with acetone, methanol, and acetonitrile. The two plates were then placed facing each other in a 100 mL beaker and connected to a Consort EV202 bench power supply. Ninety milliliters of silver trifluoroacetate solution (98%; Trimen Chemicals, Łódź, Poland) was poured into a beaker. The solution was prepared by dissolving 9 × 10^−5^ moles of AgTFA in a mixture of 80% isopropanol and 20% acetonitrile. Electrodeposition was carried out for 15 min at a voltage of 10V. The negative electrode was then cleaned with cotton wool and washed three times in boiling acetonitrile and isopropanol. LDI TOF MS measurements were performed in the *m*/*z* 80–2000 range with an UltrafleXtreme mass spectrometer (Bruker Daltonics, Bremen, Germany) equipped with a 355 nm laser with a frequency of 2 kHz. The number of laser shots was 2000 (4 × 500) per sample. The electrode voltages were 26.64 and 13.54 kV. At the first acceleration, the voltage was 25.08 kV, and for the second ion source it was 22.43 kV. The detector gain value for the reflectron was 30×. Cubic calibration was performed using silver cluster signals. An analysis of the data obtained was performed using the FlexAnalysis 3.3 software (Bruker Daltonics, Bremen, Germany) and mMass 5.5.0.

### 2.9. Analysis of B Vitamins and Lactic Acid

B vitamins: Bacterial biomass was added via a sterile loop into sterile tubes containing niacin test medium and MRS broth to obtain a 0.5 and 1 McF suspension. The cultures prepared in this way were incubated at 20 and 30 °C for 24 h. After this, the contents of the tubes were vortexed and then centrifuged (4000 rpm, 10 min at 4 °C). The obtained supernatant was used for HPLC analysis to determine B vitamin content (riboflavin—B2; niacin—B3; and folic acid—B9). A chromatographic analysis was performed using LC-MS 8050 and a Kinetex C8 column (100 × 2.1 mm, 1.7 µm). The mobile phase was a mixture of MeOH and 0.1% HCOOH in H_2_O in a gradient elution (0.01–5 min, 20–35% B; 5–6 min, 35–20% B; 6–7 min, 20% B), the flow was 0.2 mL/min, the oven temperature was 30 °C, and the injection was 3 µL. 

Lactose (L) and lactic acid (LA): Sufficient bacterial biomass was added to sterile tubes containing LAPTg medium to obtain a suspension concentration of 1 McF. The cultures prepared in this way were incubated at 20/30 °C for 24 h. After this, the contents of the tubes were vortexed and then centrifuged (4000 rpm, 10 min at 4 °C). The obtained supernatant was used for the HPLC analysis of lactose and lactic acid. Chromatographic analysis was performed using LC-MS 8050 and a Kinetex C8 column (100 × 2.1 mm, 1.7 µm). The mobile phase was 0.1% HCOOH in H_2_O with a flow rate of 0.2 mL/min, oven temperature of 30 °C, and injection of 2 µL.

In order to perform the quantitative analysis, the MRM MS (multiple reaction monitoring) scanning mode was used (Table 1). The method was validated via the use of a series of standard solutions for each of the determined compounds. For the determination of vitamins on an analytical balance, 1 mg of vitamins B2, B3, and B9 were weighed and dissolved in 1 mL of distilled water. In turn, 2.5 mg of lactic acid and 1 mg of lactose were dissolved in 1 mL of 0.1% HCOOH in H_2_O. Dilutions were prepared from the stock solutions and used for method validation (Table 2). 

### 2.10. Inhibition of Pathogens

The antimicrobial activity of the selected LAB strains (*Lactococcus lactis* and *Weissella cibaria*) was tested against 3 strains of pathogenic bacteria: *Escherichia coli*, *Enterobacter cloacae*, *Pseudomonas aeruginosa*, and one food spoilage microorganism—*Listeria innocua*. All strains came from our own collection and were isolated from clinical specimens (diabetic foot swabs) or environmental specimens. In this study, we used the supernatant of LAB strains in liquid MRS broth medium (Sigma Aldrich, Germany). For this purpose, LAB cell biomass was added to the medium such that the optical density of the suspension was about 4 McF. The samples were then incubated for 24 h at 37 °C (aerobic conditions). In order to obtain a fluid containing no bacterial cells, after incubation, the samples were centrifuged (30 min, 4000 rpm), and the obtained supernatant was used for further studies. Mueller–Hinton agar (MH; Sigma Aldrich, Steinheim, Germany) was used in this assay. MH agar is a non-selective medium that is recommended for testing the sensitivity of bacteria to various substances, e.g., antibiotics, bacteriocins, etc. The assay consisted of spreading 100 μL of the pathogen suspension on the surface of the plate. Then, after 15 min, 0.3 mL of lactic acid bacteria supernatant was applied to the surface of the medium and incubated for 24 h at 25 °C (aerobic conditions). All the tests were repeated three times for each sample.

## 3. Results and Discussion

### 3.1. Isolation of Lactic Acid Bacteria—Identification and Phenotypic Characterization

Fermented vegetables are traditionally obtained through the spontaneous fermentation of lactic acid, in which a variety of microorganisms are involved. The microflora of vegetables is varied and depends on the type of plant and the part of it being processed. Its composition significantly affects the quality and safety of final products [20]. With the use of the MALDI technique, five Gram-positive bacteria (*Lactococcus garvieae*, *Lactococcus lactis*, *Lactobacillus plantarum*, *Leuconostoc citreum*, and *Weissella cibaria*) and two fungi (*Geotrichum candidum* and *Pichia kluyveri*) were identified in a pickled red beet sample. All of the isolated bacteria were lactic acid bacteria (LAB). Two isolated strains (*W. cibaria* and *L. citereum*) were heterofermentative bacteria, which play a role in the initial steps of vegetable fermentation. In turn, the remaining bacterial strains were homofermentative bacteria, which have a higher tolerance to high salt concentrations and acidification, meaning they play a role in the final stages of the fermentation process [21]. Two LAB strains—*Lactococcus lactis* and *Weissella cibaria*—were selected for further research. *L. lactis* is a well-known LAB strain that shows high potential as a probiotic; in addition, its use in food production has a long history. Nevertheless, *L. lactis* is mainly isolated from dairy products. In our work, we focused on another type of potential probiotic source, namely silage. The second strain, *W. cibaria*, has been found to have probiotic potential and various beneficial properties, including the functional production of exopolysaccharides and attenuation of pathogen-induced inflammatory responses. In addition, this strain was found to be the dominant species in the microflora of silage products, such as kimchi [22]. One of the requirements set for starter cultures strains is their precise characterization. For this reason, additional identification was performed for these strains by sequencing the 16S rRNA gene. This gene has been the basis of sequence-based bacterial analysis for decades because it is highly conservative and present in all bacteria [23]. A comparison of the identification results obtained using the MALDI and 16S rRNA techniques is presented in Table 3.

The microscopic observation of the cell morphology of the selected bacterial strains showed that *L. lactis* and *W. cibaria* are Gram-positive bacteria (Figure 1). The *L. lactis* cells demonstrated a spherical shape (Ø 1 µm) and occurred in chains (see Figure 1A), while *W. cibaria* had rod-shaped cells growing in pairs (2.0–2.5 µm long and 0.9–1.0 µm wide) (see Figure 1B), which is consistent with their taxonomic affiliation. The selected LAB strains were tested for their ability to: (1) produce catalase; (2) produce organic acids by metabolizing glucose; (3) metabolize citrate; and (4) produce the nitrate reductase enzyme. For both strains, the results of the tests mentioned above were negative, i.e., no activity in the examined aspect.

### 3.2. Growth Assessment of Selected Isolates

The ability of the selected LAB to grow under various culture conditions was assessed by measuring optical density (OD). Measuring the OD at 565 nm is the most common method for estimating the concentration of cells in a suspension liquid [24]. For the growth assessment of *L. lactis* and *W. cibaria*, four different culture media (MRSB, TSB, LAPTg and M17), two temperatures (20 °C and 30 °C), and three measurement points (6, 24, and 48 h) were selected. During the measurements, the effect of the background absorbance by the pure medium was taken into account (MRS—1.1; TSB—0.28; M-17—6.63; LAPTg—0.68). For all strains, an increase in optical density was observed with increasing cultivation time. The degree of growth depended on the medium used and the incubation temperature. 

The results of the growth assessment showed that the six-hour incubation did not cause any significant differences between the set cultivation conditions, and the OD of both strains was comparable (0.53–0.60 McF); however, after 24 h, the differences began to become noticeable (Figure 2). Considering the lower temperature (20 °C), the growth of *L. lactis* was significantly improved when LAPTg or TSB medium was applied, and only a slight increase in OD between 24 and 48 h was observed. Similar observations were noted for *W. cibaria* after 1 day of incubation; however, the differences between these two media and the other two media were less than in the case of the former strain. Moreover, after 48 h, the best growth of *W. cibaria* was observed in MRSB, as shown by a considerable increase in the bacterial OD between 1 and 2 days of incubation—from 2.17 to 7.97 McF. Regarding the higher incubation temperature (30 °C), *W. cibaria* demonstrated a considerably higher growth in biomass after 24 and 48 hrs of incubation when MRSB was applied −1.5–5.9 times higher compared to the other culture media. Contrary to this, the growth of *L. lactis* was similar on all tested culture media except for the M17 medium, where the increase in biomass was significantly lower—by 2.2–3.2 times. In general, the M17 medium appeared to be the worst given the growth of both tested LAB strains.

### 3.3. Proteolytic Properties 

From an industrial point of view, proteolytic activity is an advantageous feature of LAB strains. In the dairy industry, proteolytic properties play a key role in ensuring successful fermentation. Proteolytic enzymes secreted by bacteria release peptides and amino acids from casein, which are involved in creating the aroma, texture, and taste of the final product [25]. The proteolytic properties of the tested strains were determined by means of a spectrophotometric test with the o-PA reagent. The concentration of FAA was calculated based on the equation of the glycine standard curve and the obtained absorbance results. The result for *L. lactis* was 2.01 ± 0.002 mmol Gly/L FFA, and for *W. cibaria* 1.91 ± 0.01 mmol Gly/L FFA (see Figure 3). 

According to data in the literature, proteolytic activity is classified as low (0–1 mmol/L), medium (1–2 mmol/L), and high (2–3 mmol/L) [25]. The proteolytic activity of the tested strains was between 1 and 2 mmol/L, which proves their intermediate activity. Joković et al. also examined the proteolytic activity of many different LAB strains isolated from dairy products using the same technique [26]. The strains of *W. cibaria* isolated by the authors were characterized with the lowest values of proteolytic activity (about 0.5 mmol Gly/L). In our work, we obtained activity at a level four times higher in the case of *W. ciabria*, which indicates the possibility of using this strain as a potential starter culture in the dairy industry. Joković et al. also tested fourteen *L. lactis* strains. As in our study, they observed that the proteolytic activity of *L. lactis* was higher than that of *W. cibaria*. In turn, the proteolytic activity of various *L. lactis* strains isolated by Gonzalez et al. during the cheese-making process ranged from 0.68 to 5.98 mM_Gly_/L [27]. In the work of Herreros et al., the range of proteolytic activity of *L. lactis* isolated from cheese was 0.22 to 2.85 mMGly/L [28]. Our results for *L. lactis* are consistent with the above values from the literature. The differences in our results may have many causes, of which the choice of environmental matrix for the isolation of the strain has the greatest influence.

### 3.4. Volatile Compounds

Lactic acid bacteria produce volatile organic compounds (VOCs) by fermenting various products. Their composition is responsible for the taste, smell, and texture of fermented products, e.g., dairy products. VOCs produced by *L. lactis* and *W. cibaria* strains were analyzed using gas chromatography (GC). A control analysis of an applied culture medium was also performed. The peaks in the chromatograms were matched with compounds in the NIST 17 library (see Figure 4). Sixteen compounds were identified in the *L. lactis* chromatogram and thirteen in *W. cibaria*. They were mainly alcohols, ketones, acids, and aldehydes. Ten of the identified compounds were present in both LAB strain samples. Six compounds were identified in the control sample. Four of them (methoxyphenyloxime, benzaldehyde, 2,2,4,6,6-pentamethylheptane, and 2-ethyl-1-hexanol) were also identified in samples of both LAB strains. In the case of the *L. lactis* strain, benzene acetic aldehyde was additionally identified, which was also present in the control sample.

LAB strains take part in three basic biochemical processes that are responsible for the basic taste of fermented milk products. They are glycolysis (the metabolism of lactose, lactate, and citrate), lipolysis, and proteolysis [29]. As a result of the metabolism of citrate or lactose, the main fragrance compound produced is acetoin, which gives a buttery (acidic, sour, cheesy, dairy, and creamy with a fruity nuance) odor [30,31]. In our research, this compound was identified only in the *L. lactis* strain; however, its amount was much higher compared to the other compounds. Interestingly, although *L. lactis* is one of the most commonly used strains in the production of fermented foods, it usually requires a chemical or genetic mutation to increase acetoin production [30]. In turn, as a result of the oxidative degradation of fatty acids and amino acids, methyl ketones and aldehydes are formed, which can be modified by dehydrogenases to form alcohols and organic acids [29]. For example, 2-nonanone, 1-nonanone, and 2-heptanone—identified in the samples—are compounds that are very common in dairy products, giving them a hot milk and dairy-like aroma and delicate taste [29,32]. 3-methyl-1-butanol, identified in both samples, and 3-methylbutanoic acid, identified in *W. cibaria*, give dairy products a characteristic cheese aroma. Butanoic acid is present in almost all dairy products (also in both of our strains) and is considered as the most potent odorant, giving a buttery, sweaty, cheesy, and rancid odor [32].

### 3.5. Analysis of Fats 

During the surface-assisted laser desorption/ionization analysis, the spectra of the tested strains were recorded. The assignment of the probable metabolites, peptides, and lipids to the *m*/*z* values was carried out by searching the HMDB [33], LIPID MAPS [34], and MetaCyc [35] databases. The compounds identified in the spectra were present in the form of adducts with a proton, sodium, and potassium, and by using a plate covered with silver nanoparticles, we also found adducts with two Ag isotopes, i.e., [M + ^107^Ag]^+^ and [M + ^109^Ag]^+^. The possibility of the silver cationization of the analyzed compounds is not surprising and has been described by many authors who have used SALDI methods based on plates with silver nanoparticles [36,37,38,39]. SALDI methods based on silver nanoparticles have an established position in the analysis of compounds with low molecular weights, such as metabolites or lipids [40]; however, it is worth noting that none of these techniques have thus far been applied to the analysis of lipids from microorganisms. 

For the analysis of the spectra, rigorous criteria were used when searching for compounds in databases: the signal-to-noise ratio was a minimum of 5, and the match error was no more than |10| ppm. In the SALDI MS spectrum of the *L. lactis* extract, 67 signals were identified that were assigned to 64 compounds (Table 4), while for *W. cibaria*, there were 84 signals matched with 77 compounds (Table 5). Despite the use of the Folch extraction method, the spectra showed many signals not only from lipids, but also from peptides. In some cases, extracts from bacterial cells may be contaminated with peptides, sugars, free amino acids, and other endogenous compounds. Peptides trapped in lipid micelles can enter the organic phase during the extraction process [41]. Lactic acid bacteria are known for their production of bioactive peptides produced from proteins by microbial proteolysis [42,43,44]; thus, it is not surprising that they were present in cell extracts. The remaining signals on the spectra were mainly assigned to lipids belonging to all classes, i.e., fatty acids, glycerolipids, glycerophospholipids, sphingolipids, and steroids. Fatty acids (FA) are one of the most dynamic components of bacterial cells. They are components of phospholipids and play a key role in maintaining the structure and function of the cell membrane [10]. The glycerolipids identified in the spectra were triacylglycerols (TGs) and diacylglycerols, which are a reserve and energy material in bacterial cells. Glycerolipids also regulate the level of lipids in cell membranes and are precursors in the synthesis of phospholipids [45,46,47]. Among the glycerophospholipids identified were phosphatidylglycerols (PGs), which are the main phospholipids among lactic bacteria; phosphatidylethanolamines (PEs), responsible for membrane architecture and bacterial mobility; phosphatidylinositols (PIs), playing a role in the dynamics of the bacterial membrane [48,49]; phosphatidylcholine (PC), which plays a key role in symbiotic interactions and resistance to bacteria and heavy metals; and phosphatidylserine (PS), which is a precursor in the synthesis of PEs [10,50]. Another group of lipids detected in the extracts were ceramides, which belong to sphingolipids. They act as a protective barrier by trapping water in cells [51]. Sterols are another class of lipids that were identified in the spectra. These compounds have been detected in bacteria; however, there is insufficient research to confirm their properties and importance [52]. Among the other compounds identified in the spectrum of the *L. lactis* extract, we distinguished celotriose (*m*/*z* 613.075), which is glucotriose, a compound that acts as a bacterial xenobiotic metabolite. Interestingly, in their work, Adsul et al. showed the possibility of using cellotriosis by Lactobacillus spp. bacteria as a carbon source and for the microbial production of lactic acid [53].

### 3.6. Synthesis of Vitamins and Lactic Acid

A growing consumer market demand for products containing functional ingredients that provide medical or health benefits has caused probiotic selection to be currently focused on finding certain strains of LAB that are able to produce or increase the content of specific beneficial compounds found in foods [54]. Such compounds could be micronutrients, for example, B-group vitamins, including riboflavin (B2), niacin (B3), and folate (B9), or substances with preservative properties, such as lactic acid. This study determined the ability of the *L. lactis* and *W. cibaria* strains to synthesize B vitamins and lactic acid. The concentration of these compounds was determined using HPLC in the selected LAB supernatants obtained after cultivation in two different culture media (niacin assay medium, and MRS broth) (Figure 5).

The results from the analysis of vitamin B (B2, B3, and B9) production capacity among the tested LABs revealed that *L. lactis* secreted niacin and riboflavin regardless of the type of culture medium used, and the amounts were highly temperature dependent (Table 6). The concentrations of secreted niacin were higher compared to that of riboflavin by 1.7 (30 °C) and 2.5 (20 °C) times and reached values of 1.273 ± 0.177 and 3.582 ± 0.132 µg/mL, respectively. The difference between niacin and riboflavin secretion levels in *L. lactis* increased up to 4.9 times when the MRS broth medium was applied; however, the measured amounts were five (niacin) and ten (riboflavin) times lower compared to those obtained using the niacin assay medium. Although *W. cibaria* did not demonstrate the ability to produce any type of the investigated B vitamins on the niacin assay medium, this changed when MRSB was applied and the secreted niacin and riboflavin amounts reached 1.465 ± 0.188 and 0.109 ± 0.002 µg/mL, respectively. Both strains revealed the high effect of the incubation temperature on the levels of secreted B vitamins. In each case, the use of a lower temperature (20 °C) stimulated vitamin production. This was particularly evident in the MRS broth medium, where lowering the temperature from 30 °C to 20 °C caused the strains to shift from consumers to vitamin B producers. Regarding folates, the analysis revealed that none of the investigated LAB were able to secrete vitamin B9 regardless of the temperature and type of culture medium used.

Our studies revealed that one of the two investigated LAB strains derived from the pickled beetroots, *L. lactis*, was able to secrete niacin and riboflavin in the CDM medium. Thus far, riboflavin-producing LAB have been isolated from different ecological niches and demonstrated great variation in their ability to increase the concentrations of this vitamin in food matrices, including milk, soymilk, whey, and pseudocereals, from 0.50 to 6.57 mg/L [55,56,57]. However, the vast majority of work on riboflavin production relates to lactobacilli species, such as *L. fermentum*, *L. acidophilus*, or *L. plantarum* [58,59]. Natural (wild types) of *Lactococcus lactis* are considered as riboflavin consumers who can be easily converted into riboflavin producers through the overexpression of riboflavin biosynthesis genes via genetic engineering and exposure to purines or a toxic riboflavin analogue (roseoflavin) [60,61]. The niacin assay medium used in our studies contains sources of purines (adenine sulfate and guanine hydrochloride) and was likely responsible for observed significant increase in riboflavin production in *L. lactis* compared to the MRS broth medium. The recorded level was as high as that within the lactobacilli group—1.409 mg/L—and considering that the recommended daily intake for riboflavin is 1.3 mg/day for men and 1.1 mg/day for women [62], isolated *L. lactis* demonstrates high potential for the fortification of foods with vitamin B2 providing that suitable culture conditions (20 °C, purine addition) are established.

Interestingly, although *Lactococcus* members are not known to produce nicotinate metabolites, to which niacin belongs, the investigated *L. lactis* strain was characterized by the highest vitamin B3 secretion among all tested B-group vitamins. In their work, Jung et al. detected niacin production for the first time in a *Lactococcus* member, namely in *L. raffinolactis* WiKim0068, isolated from fermented cabbage (kimchi) [63]. The detected niacin concentration (0.932 mg/L) was similar to that obtained for *Leuconostoc* spp. (0.837–1.05 mg/L) and was far higher when *Lactobacillus* species were applied −0.05–0.1 mg/L. The niacin production capacity of *L. raffinolactis* WiKim0068 was slightly higher compared to that of *L. lactis* cultured on the same medium (MRSB) in our study; however, such differences may have resulted from varied incubation times, i.e., 48 h compared to 24 h, respectively. Moreover, as we showed, the niacin secretion level of *L. lactis* was significantly increased when vitamin-free medium combined with a lower temperature (20 °C) was applied. Another strain investigated in our studies, *Weissella cibaria*, is believed to be an interesting probiotic candidate that is already being used as a commercially available oral care probiotic in Korea (*W. cibaria* CMU) [64] and has been shown to produce B-group vitamins (e.g., folic acid [65]). Although the *W. cibaria* strain did not secrete any of the investigated B vitamins under most of the culture conditions used, the application of MRS broth medium combined with lower temperature brought about a switch from consumption to vitamin B production. Nevertheless, in most cases, *W. cibaria* appeared to be auxotrophic for riboflavin, since its initial concentration after incubation significantly decreased. As both strains investigated in our studies originated from the same source (pickled beetroot), it is possible that the sharing of the same ecological niche by *L. lactis* and *W. cibaria* has conditioned riboflavin production by the former strain that further can be consumed by the latter.

Similar to vitamin B production, the investigated LABs demonstrated great differences in their ability to utilize lactose and produce lactic acid (Table 7). At the lower temperature, *L. lactis* addition caused a 93% decrease in the lactose concentration of the medium, while in the presence of *W. cibaria* it was decreased only by 40%. The opposite phenomenon was observed at a higher temperature, wherein W. cibaria metabolized almost all lactose present in the culture medium (98%); in the case of L. lactis, this value was 67%. Regardless of the temperature and lactose metabolism rate, the highest lactic acid production was noted for variants with *L. lactis* inoculation, in which it was 2.3–3.3 times higher than that of *W. cibaria*. The lactic acid concentration produced by *L. lactis* was 1014.99 ± 47.59 µg/mL at 20 °C and 638.21 ± 8.03 µg/mL at 30 °C. Despite the great variation in lactose utilization by *W. cibaria* depending on the applied temperature, the differences in the level of lactic acid production under different incubation conditions were only slight—310.99 ± 7.48 and 277.36 ± 5.82 µg/mL at 20 °C and 30 °C, respectively.

The main product of the fermentation process carried out by LAB is lactic acid (LA), which is authorized by the U.S. Food and Drug Administration as GRAS (generally regarded as safe). Differences between the main types of LAB may explain the observed variation in lactose metabolism and lactic acid production between the investigated strains, since L. lactis is a homofermentative LAB while *W. cibaria* is a heterofermentative LAB. Indeed, the greater consumption of lactose by the *L. lactis* strain was accompanied by a greater production of lactic acid, while in the case of *W. cibaria*, despite the large differences in the percentage of metabolized lactose depending on the temperature used, the amount of lactic acid secreted was very similar. This suggests that in the case of *W. cibaria*, which demonstrated high lactose consumption under 30 °C, raising the temperature primarily causes an increase in the secretion of fermentation products other than LA, e.g., acetate or ethanol, which were already detected in the *W. cibaria* strain 92 previously [66]. *L. lactis* was found to be far more effective than *W. cibaria* at the decreasing the lactose concentration and LA production when the lower temperature was applied; a 93% decrease in lactose was accompanied by LA production at a level of 1 g/L, while for the second strain it was only a 40% decrease and the LA level was 0.310 g/L. These findings agree with the common sentiment that only homofermentative LAB is available for commercial LA production due to the high yield, productivity, and a high optical purity of lactic acid [8]. Although lactic acid is especially in demand on the market due to its many applications in the food industry as a food preservative, fermentation agent, acidulant, flavour enhancer, etc., other products of fermentation carried out by LAB can also be beneficial, e.g., acetate can serve as a nutrition source for other microorganisms in the colon, including propionate and butyrate-producing microorganisms [67]. In light of this, both tested strains demonstrated the desired probiotic properties, showing that fermented plant products can serve as valuable potential probiotic sources. Furthermore, during the evaluation of potential probiotic sources, it is crucially important to check their activity under different culture conditions, including different temperatures and culture media compositions, as this will help in planning their future application.

### 3.7. Study of Antagonistic Abilities

One of the criteria for selecting a good probiotic or starter culture strain is its ability to enhance the host’s natural defense against food pathogens. The reason for the inhibition of the growth of pathogenic bacteria is most likely the presence of organic acids, hydrogen peroxide, and bacteriocins in the culture fluid. Bacteriocins are proteins or peptides produced by bacteria that exhibit a bacteriostatic or bactericidal activity that is specific to some organisms [68]. In order to investigate the antibacterial properties of the isolated LAB strains, we used three pathogenic strains and one food spoilage strain. The general effect of the LAB supernatant on the growth of the selected bacteria on the medium was checked. Table 8 shows the average colony numbers of the pathogenic bacteria without and with the addition of the LAB supernatant. In order to better visualize the reduction in the number of pathogenic colonies, the numerical values were converted into percentage values. Similar antimicrobial activity was observed for both strains, *W. cibaria* and *L. lactis*. Both strains completely inhibited the growth of *P. aeruginosa* and *L. innocua.* They inhibited the growth of *E. coli* and *B. cereus* to a similar extent. In the case of *E. cloacae*, the *W. cibaria* strain showed greater antimicrobial properties. 

Two types of bacteriocins have been identified in lactic acid bactera, lanthibiotics (characterized by the presence of dehydrated and/or thioether amino acids) and non-lanthibiotics (those containing unmodified amino acids) [69]. *Lactococcus lactis* is a very popular component of commercial starter cultures applied by the dairy industry; thus, its bacteriocins have been widely characterised. The first bacteriocin isolated from *L. lactis* was nisin, which can kill a wide range of Gram-positive bacteria. For this reason, it is widely used as a food preservative in processed cheeses and dairy products [70]. *L. lactis* can also produce other lanthibiotics, such as the single peptide lacticin 481 and the two-component system lacticin 3147, and non-lanthibiotics, including lactococcin MMFII and lactococcin B or G [69]. By reviewing the literature on the antagonistic properties of the *Weissella cibaria* strain, one can find confirmation of the results obtained during our research. Srionnual et al. focused on the characterization of the bacteriocins responsible for the antagonistic properties of this strain [71]. They showed that this strain produced a bacteriocin, which they named Weissellicin 110. It was stable after high temperature treatment but had a narrow inhibitory spectrum against *Listeria monocytogenes*. Their research also indicated that the optimal temperature for the production of bacteriocins by *W. cibaria* was around 30 °C. In our research, the *W. cibaria* isolated from beetroot showed the strongest properties against bacteria of the same genus (*Listeria innocua* in our research). Kant Lakra et al. also showed in their research that the *Weissella cibaria* MD2 strain had potential probiotic properties [72]. The strain showed high survival in the artificial environment of the gastrointestinal tract, adhesion to intestinal epithelial cells, and antimicrobial activity against food-borne pathogens. They indicated that the antimicrobial properties of the strain could be used for packaging (as edible films) in the food industry in order to extend shelf-life. The strain they isolated also showed pro-health effects, such as high cholesterol removal and antioxidant activity, as well as a lack of haemolytic activity and sensitivity to conventional antibiotics, which makes it safe to use as a starter culture in the food industry or for the preparation of new functional foods.

## 4. Conclusions

Although a number of probiotic strains have been isolated and characterized, the search for more effective strains is still ongoing. The most attractive source of new bacteria with potential probiotic properties is fermented food, including popular dairy products, as well as silage, which is becoming more and more popular. In this study, two LAB strains—*Lactococcus lactis* and *Weissella cibaria*—isolated from a pickled beetroot sample were identified and characterized. During the analyses, two aspects were taken into account: their potential probiotic properties and their applicability in the food industry. Both lactic acid bacteria showed antimicrobial activity against four pathogenic bacteria (*Escherichia coli*, *Enterobacter cloacae*, *Pseudomonas aeruginosa,* and *Listeria innocua*). However, *L. lactis* showed much more interesting properties from the point of view of the food industry. The proteolytic activity of *L. lactis* was higher than that of *W. cibaria*. This strain produced 2.3–3.3 times more lactic acid than *W. cibaria* and secreted niacin and riboflavin regardless of the type of culture medium used, and the amounts were highly temperature dependent. Moreover, the analysis of volatile compounds showed that *L. lactis* produces large amounts of acetoin, which gives dairy products a desirable buttery odor. The lipid profile of the tested strains was also characterized using the modern SALDI technique, which allowed for the detection of lipids belonging to all classes, i.e., fatty acids, glycerolipids, glycerophospholipids, sphingolipids, and steroids.

Both tested strains demonstrated desirable probiotic properties, showing that fermented plant products can serve as valuable potential probiotic sources. Furthermore, during the evaluation of potential probiotic sources, it is crucially important to check their activity under different culture conditions, including temperatures and culture media compositions, to help plan their future applications.

## Figures and Tables

**Figure 1 foods-11-02257-f001:**
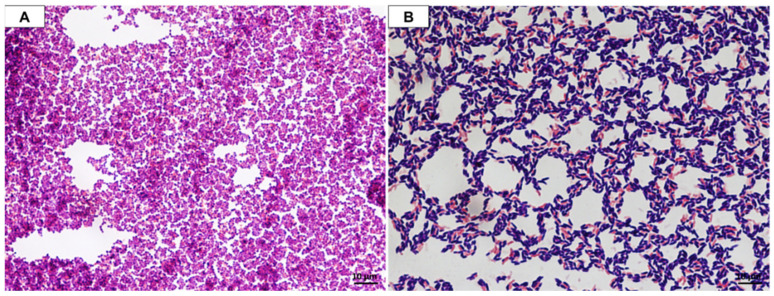
Optical microscope photos of the Gram staining of the tested strains: (**A**)—*Lactococcus lactis*; (**B**)—*Weissella cibaria*.

**Figure 2 foods-11-02257-f002:**
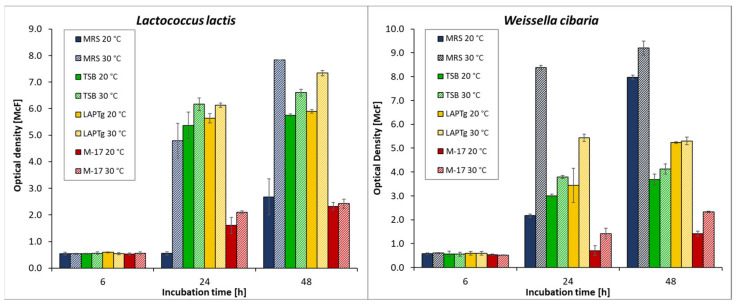
Graphs showing the growth of the tested LAB strains in various culture media and temperatures.

**Figure 3 foods-11-02257-f003:**
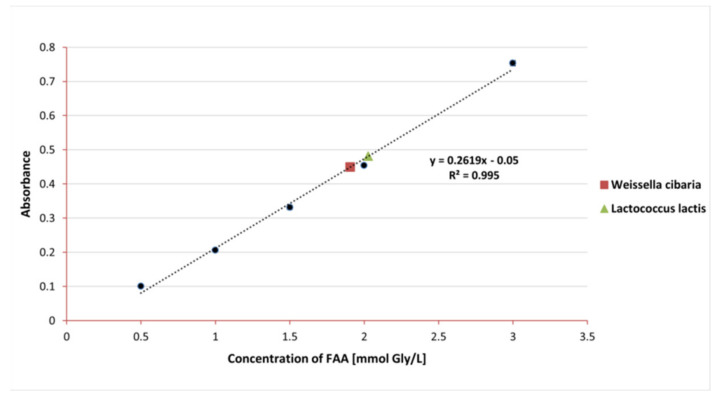
Calibration curve showing the dependence of absorbance on the concentration of glycine.

**Figure 4 foods-11-02257-f004:**
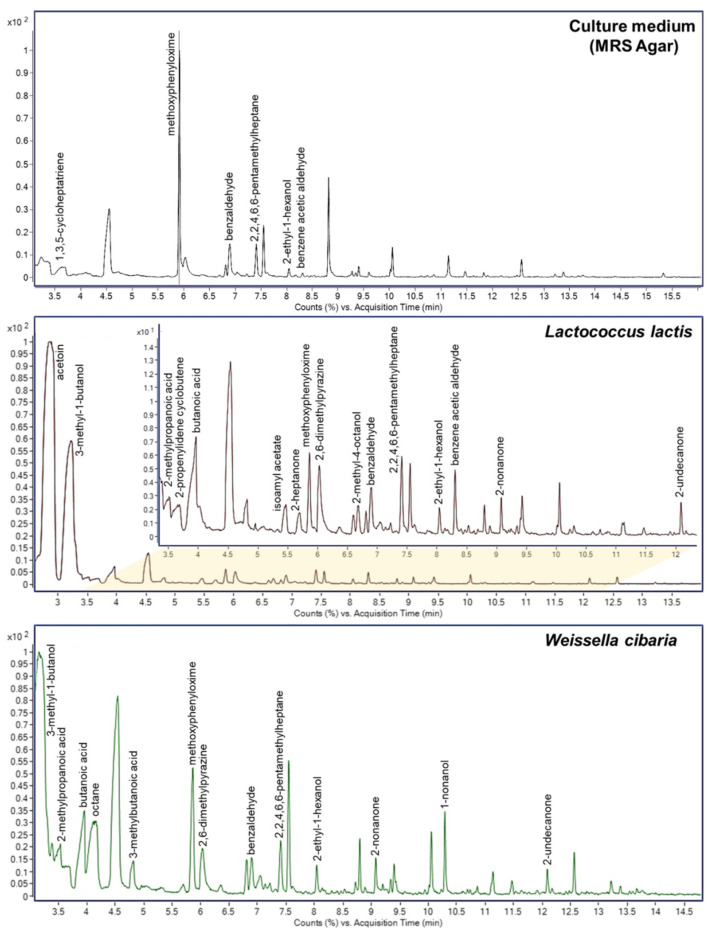
GC/MS chromatograms of control sample (MRS agar) and analyzed LAB strain samples.

**Figure 5 foods-11-02257-f005:**
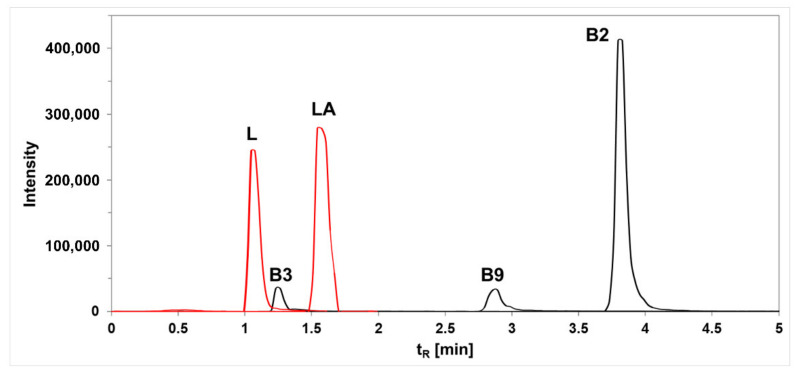
Exemplary chromatograms obtained during the analysis of selected compounds by HPLC. L—lactose; LA—lactic acid; B2—riboflavin; B3—niacin; B9—folic acid; tR—retention time.

**Table 1 foods-11-02257-t001:** Conditions for the detection of the analyzed compounds. B2—riboflavin; B3—niacin; B9—folic acid; L—lactose; LA—lactic acid.

	Retention Time (Min)	Ionization	Molecular Ion	Fragment Ion	Q1 (V)	CE	Q3 (V)
**B2**	3.811	Positive	337	243	−20	−25	−20
**B3**	1.247	Positive	124	79	−10	−24	−15
**B9**	2.971	Positive	442	176	−16	−39	−19
**L**	1.060	Negative	341	161.15	13	8	15
**LA**	1.594	Negative	89.25	43.05	19	13	14

**Table 2 foods-11-02257-t002:** Validation of the analytical method. LOD—limit of detection; LOQ—limit of quantification.

	Linearity (µg/mL)	R^2^	Standard Curve Equation	LOD (µg/mL)	LOQ (µg/mL)
**B2**	0.005–10	0.9941	y = 1,137,260x − 89,427	0.001	0.0033
**B3**	0.25–10	0.9985	y = 8819.9x − 1000.03	0.050	0.1650
**B9**	0.025–10	0.9980	y = 104,830x + 11,539	0.005	0.0165
**L**	0.01–400	0.9963	y = 19,947x + 122,904	0.005	0.0165
**LA**	0.01–2000	0.9972	y = 18,770x + 651,622	0.005	0.0165

**Table 3 foods-11-02257-t003:** Comparison of identification results between the MALDI and 16S rRNA techniques.

Identification by MALDI		16S rRNA Identification			Number in PCM **
Identification Result	MSP Log Score *	Identification Result	% Similarity	Strand Length (bp)	
*Lactococcus lactis ssp lactis* 3C1_QSA IBS	2.02	*Lactococcus lactis* strain NBRC 100933 [NR_113960.1] *Lactococcus lactis* strain NCDO 604 [NR_040955.1]	99.86 99.86	1412	*L. lactis* Beet 1 [OM957496]
*Weissella cibaria* DSM 15878T DSM	2.13	*Weissella cibaria* strain II-I-59 [NR_036924.1] *Weissella confusa* strain JCM 1093 [NR_040816.1]	99.58 99.17	1442	*W. cibaria* Beet 2 [OM957500]

* log score >2.00—secure genus identification, probable species identification. ** PCM—Polish Collection of Microorganisms.

**Table 4 foods-11-02257-t004:** SALDI mass spectrum of *Lactococcus lactis* lipid extract made on an AgNLET plate with electroplated silver nanoparticles (S/N ≥ 5, Δ max. ± 10 ppm).

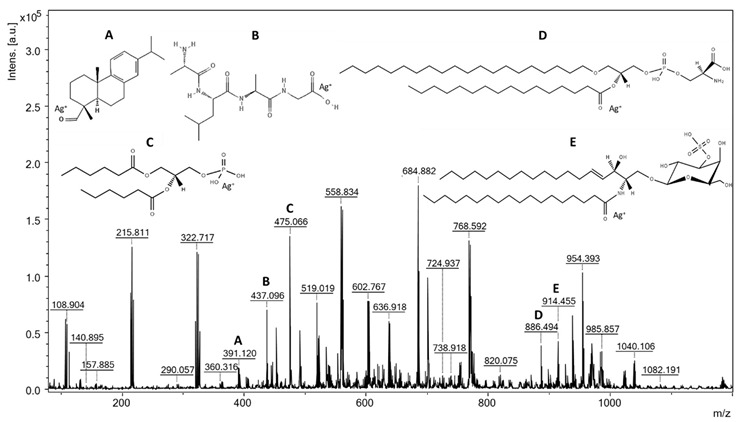					
	Compound *	Adduct Formula	*m*/*z* Exp.	*m*/*z* Calc.	Δ (ppm)
**1.**	[FA oxo(15:0)] 4-oxo-pentadecanoic acid	[C_15_H_28_O_3_ + ^107^Ag]^+^	363.108	363.108	0.0
**2.**	[FA hydroxy(17:2)]12-hydroxy-8E,10Eheptadecadienoic acid	[C_17_H_30_O_3_ + ^107^Ag]^+^	389.123	389.124	−2.6
**3.**	Dehydroabietadienal	[C_20_H_28_O + ^107^Ag]^+^	391.120	391.119	2.6
		[C_20_H_28_O + ^109^Ag]^+^	393.119	393.118	2.5
**4.**	[GP (6:0/6:0)] 1,2-dihexanoyl-sn-glycero-3-phosphate	[C_15_H_29_O_8_P + K]^+^	407.123	407.123	0.0
**5.**	Ala-Leu-Ala-Gly	[C_14_H_26_N_4_O_5_ + ^107^Ag]^+^	437.096	437.095	2.3
**6.**	Met-Phe-Cys	[C_17_H_25_N_3_O_4_S_2_ + K]^+^	438.093	438.092	2.3
**7.**	Cys-Trp-Gly-Gly	[C_18_H_23_N_5_O_5_S + Na]^+^	444.130	444.131	−2.3
**8.**	Asn-Thr-Cys-Ser	[C_14_H_25_N_5_O_8_S + Na]^+^	446.132	446.132	0.0
**9.**	Ala-Ala-Ala-Asp	[C_13_H_22_N_4_O_7_ + ^107^Ag]^+^	453.052	453.053	−2.2
**10.**	Ala-Asn-Gly-Ser	[C_12_H_21_N_5_O_7_ + ^107^Ag]^+^	454.050	454.049	2.2
**11.**	[GP (6:0/6:0)] 1,2-dihexanoyl-sn-glycero-3-phosphate	[C_15_H_29_O_8_P + ^107^Ag]^+^	475.066	475.065	2.1
**12.**	Val-Asp-His	[C_15_H_23_N_5_O_6_ + ^107^Ag]^+^	476.069	476.069	0.0
**13.**	[GP (6:0/6:0)] 1,2-dihexanoyl-sn-glycero-3-phosphate	[C_15_H_29_O_8_P + ^109^Ag]^+^	477.065	477.064	2.1
**14.**	PA(17:2(9Z,12Z)/0:0)	[C_20_H_37_O_7_P + ^109^Ag]^+^	529.130	529.132	−3.8
**15.**	Ala-Trp-Val-Tyr	[C_28_H_35_N_5_O_6_ + H]^+^	538.267	538.266	1.9
**16.**	C12-ACP (Dodecanoyl-ACP)	[C_23_H_44_N_2_O_8_PS + H]^+^	540.263	540.263	0.0
**17.**	LysoPE(20:3(8Z,11Z,14Z)/0:0)	[C_25_H_46_NO_7_P + K]^+^	542.259	542.264	−9.2
**18.**	Asp-Leu-Met-His	[C_21_H_34_N_6_O_7_S + K]^+^	553.184	553.184	0.0
**19.**	Alpha-Trisaccharide	[C_20_H_37_NO_14_ + K]+	554.187	554.185	3.6
**20.**	Ala-Ala-Tyr-His	[C_21_H_28_N_6_O_6_ + ^109^Ag]^+^	569.110	569.111	−1.8
**21.**	Glu-Phe-Ala-Pro	[C_22_H_30_N_4_O_7_ + ^109^Ag]^+^	571.116	571.116	0.0
**22.**	Cys-Pro-Pro-Tyr	[C_22_H_30_N_4_O_6_S + ^107^Ag]^+^	585.094	585.093	1.7
**23.**	Ala-Cys-Tyr-His	[C_21_H_28_N_6_O_6_S + ^107^Ag]^+^	599.084	599.084	0.0
		[C_21_H_28_N_6_O_6_S + ^109^Ag]^+^	601.085	601.083	3.3
**24.**	Cellotriose	[C_18_H_32_O_16_ + ^109^Ag]^+^	613.075	613.073	3.3
**25.**	Octacosanyl hexadecanoate	[C_44_H_88_O_2_ + H]^+^	649.684	649.686	−3.1
**26.**	FAD stem group	[C_15_H_26_N_6_O_13_P_2_ + ^109^Ag]^+^	669.009	669.008	1.5
**27.**	GalCer(d18:2/20:0)	[C_44_H_83_NO_8_ + H]^+^	754.621	754.619	2.7
**28.**	TG(13:0/13:0/17:2(9Z,12Z))	[C_46_H_84_O_6_ + Na]^+^	755.614	755.616	−2.6
**29.**	PC(18:2(9Z,12Z)/P-18:1(11Z))	[C_44_H_82_NO_7_P + H]^+^	768.592	768.590	2.6
**30.**	[PC (16:2/18:1)] 1-hexadecyl-2-(9Z-octadecenyl)sn- glycero-3-phosphocholine	[C_42_H_86_NO_6_P + K]^+^	770.587	770.582	6.5
**31.**	PA(19:0/22:2(13Z,16Z))	[C_44_H_83_O_8_P + H]^+^	771.584	771.590	−7.8
**32.**	PC(15:0/20:2(11Z,14Z))	[C_43_H_82_NO_8_P + H]^+^	772.582	772.585	−3.9
**33.**	PG(P-20:0/17:2(9Z,12Z))	[C_43_H_81_O_9_P + H]^+^	773.572	773.569	3.9
**34.**	PC(14:1(9Z)/18:3(6Z,9Z,12Z))	[C_40_H_72_NO_8_P + ^109^Ag]^+^	834.407	834.404	3.6
**35.**	[PE (6:0/8:0)] 1-(6-[5]-ladderane-hexanyl)-2(8-[3]- ladderane-octanyl)-sn-glycerophosphoethanolamine	[C_43_H_72_NO_6_P + ^107^Ag]^+^	836.407	836.414	−8.4
**36.**	[PC (18:1/22:6)] 1-(11Z-octadecenoyl)-2-(4Z,7Z,10Z,13Z,16Z,19Z-docosahexaenoyl)-sn-glycero-3-phosphocholine	[C_48_H_82_NO_8_P + K]^+^	870.543	870.541	2.3
**37.**	PI(16:1(9Z)/19:1(9Z))	[C_44_H_81_O_13_P + Na]^+^	871.535	871.531	4.6
**38.**	PS(O-20:0/16:0)	[C_42_H_84_NO_9_P + ^109^Ag]^+^	886.494	886.493	1.1
**39.**	PI(15:0/20:2(11Z,14Z))	[C_44_H_81_O_13_P + K]^+^	887.497	887.505	−9.0
**40.**	PE(P-18:0/22:4(7Z,10Z,13Z,16Z))	[C_45_H_82_NO_7_P + ^109^Ag]^+^	888.495	888.487	9.0
**41.**	PS(O-16:0/21:0)	[C_43_H_86_NO_9_P + ^107^Ag]^+^	898.505	898.509	−4.5
**42.**	PI(16:0/20:3(8Z,11Z,14Z))	[C_45_H_81_O_13_P + K]^+^	899.504	899.505	−1.1
**43.**	(3′-sulfo)Galbeta-Cer(d18:1/2-OH-16:0)	[C_40_H_77_NO_12_S + ^107^Ag]^+^	902.422	902.421	1.1
**44.**	PG(16:1(9Z)/22:4(7Z,10Z,13Z,16Z))	[C_44_H_77_O_10_P + ^107^Ag]^+^	903.428	903.430	−2.2
**45.**	PE(22:1(13Z)/24:1(15Z))	[C_51_H_98_NO_8_P + Na]^+^	906.688	906.692	−4.4
**46.**	PS(15:0/22:1(11Z))	[C_43_H_82_NO_10_P + ^107^Ag]^+^	910.468	910.472	−4.4
**47.**	PC(18:3(6Z,9Z,12Z)/20:4(5Z,8Z,11Z,14Z))	[C_46_H_78_NO_8_P + ^109^Ag]^+^	912.451	912.451	0.0
**48.**	PG(P-18:0/22:6(4Z,7Z,10Z,13Z,16Z,19Z))	[C_46_H_79_O_9_P + ^107^Ag]^+^	913.450	913.451	−1.1
**49.**	(3′-sulfo)GalBeta-Cer(d18:1/18:0)	[C_42_H_81_NO_11_S + ^107^Ag]^+^	914.455	914.458	−3.3
**50.**	PI(P-16:0/17:0)	[C_42_H_81_O_12_P + ^107^Ag]^+^	915.455	915.451	4.4
**51.**	(3′-sulfo)GalBeta-Cer(d18:1/18:0)	[C_42_H_81_NO_11_S + ^109^Ag]^+^	916.463	916.457	6.5
**52.**	PI(O-16:0/18:3(9Z,12Z,15Z))	[C_43_H_79_O_12_P + ^109^Ag]^+^	927.437	927.435	2.2
**53.**	PS(17:2(9Z,12Z)/22:4(7Z,10Z,13Z,16Z))	[C_45_H_76_NO_10_P + ^107^Ag]^+^	928.421	928.425	−4.3
**54.**	PS(18:1(9Z)/22:6(4Z,7Z,10Z,13Z,16Z,19Z))	[C_46_H_76_NO_10_P + ^107^Ag]^+^	940.430	940.425	5.3
**55.**	PI(16:1(9Z)/18:1(9Z))	[C_43_H_79_O_13_P + ^107^Ag]^+^	941.432	941.430	2.1
**56.**	[PE (22:6/22:6)] 1,2-di-(4Z,7Z,10Z,13Z,16Z,19Z-docosahexaenoyl)-sn-glycero-3-phosphoethanolamine	[C_49_H_74_NO_8_P + ^107^Ag]^+^	942.413	942.420	−7.4
**57.**	GalCer(d18:0/26:1)	[C_50_H_97_NO_8_ + ^109^Ag]^+^	948.626	948.626	0.0
**58.**	[GL (18:0/20:0/20:0)] 1-octadecanoyl-2,3-dieicosanoyl-sn-glycerol	[C_61_H_118_O_6_ + K]^+^	985.857	985.856	1.01
**59.**	[PC (24:0/26:0)] 1-tetracosanoyl-2-hexacosanoyl-sn- glycero-3-phosphocholine	[C_58_H_116_NO_8_P + H]^+^	986.859	986.851	8.1
**60.**	PE(22:2(13Z,16Z)/24:1(15Z))	[C_51_H_96_NO_8_P + ^109^Ag]^+^	990.593	990.592	1.0
**61.**	[GP (18:0/18:0/2:0/2:0)] 1,2-di-(9Z-octadecenoyl)-sn- glycero-3-cytidine-5′-diphosphate	[C_48_H_85_N_3_O_15_P_2_ + H]^+^	1006.555	1006.552	3.0
**62.**	Siroheme amide	[C_42_H_47_FeN_5_O_15_ + ^107^Ag]^+^	1024.151	1024.146	4.9
**63.**	2-Oxo-delta3-4,5,5-trimethylcyclopentenylacetyl-CoA	[C_31_H_48_N_7_O_18_P_3_S + ^109^Ag]^+^	1040.106	1040.103	2.9
**64.**	CoA(22:6(4Z,7Z,10Z,13Z,16Z,19Z))	[C_43_H_66_N_7_O_17_P_3_S + ^107^Ag]^+^	1184.255	1184.249	5.1
		[C_43_H_66_N_7_O_17_P_3_S + ^109^Ag]^+^	1186.246	1186.249	−2.5

* Putative metabolite.

**Table 5 foods-11-02257-t005:** SALDI mass spectrum of *Weissella cibaria* lipid extract made on an AgNLET plate with electroplated silver nanoparticles (S/N ≥ 5, Δ max. ± 10 ppm).

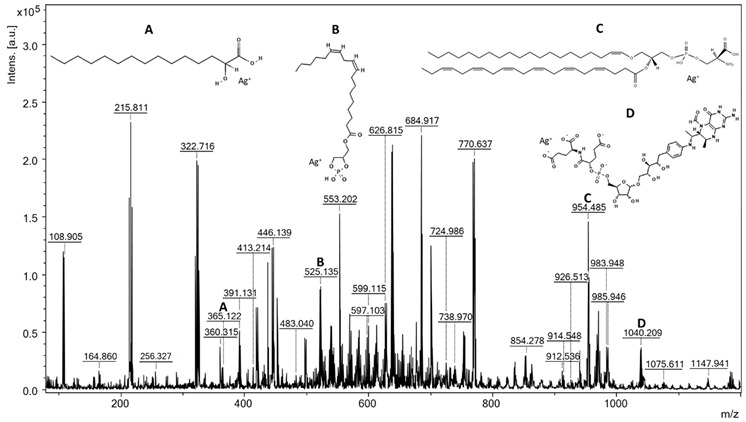					
	Compound *	Adduct Formula	*m*/*z* Exp.	*m*/*z* Calc.	Δ (ppm)
**1.**	[FA hydroxy(15:0)] 2-hydroxy-pentadecanoic acid	[C_15_H_30_O_3_ + ^107^Ag]^+^	365.12	365.12	0.0
**2.**	N-palmitoyl proline	[C_21_H_39_NO_3_ + Na]^+^	376.28	376.28	0.0
**3.**	Galactan	[C_14_H_26_O_11_ + Na]^+^	393.14	393.14	0.0
**4.**	[PC (6:0)] 1-hexanoyl-sn-glycero3-phosphocholine	[C_14_H_30_NO_7_P + K]^+^	394.14	394.14	0.0
**5.**	2-hydroxy-9Z,12Z,15Z-octadecatrienoic acid	[C_18_H_30_O_3_ + ^109^Ag]^+^	403.12	403.12	0.0
**6.**	Met-Tyr	[C_14_H_20_N_2_O_4_S + ^107^Ag]^+^	419.02	419.02	0.0
**7.**	[PC (3:0)] 1-(2E-propionyl)sn-glycero-3- phosphocholine	[C_11_H_22_NO_7_P + ^109^Ag]^+^	420.02	420.02	0.0
**8.**	S-(2-Chloroacetyl)glutathione	[C_12_H_18_N_3_O_7_ClS + K]^+^	422.02	422.02	0.0
**9.**	Val-Gly-Arg	[C_13_H_26_N_6_O_4_ + ^107^Ag]^+^	437.11	437.11	0.0
**10.**	S-Adenosylmethionine	[C_15_H_23_N_6_O_5_S + K]^+^	438.11	438.11	0.0
**11.**	Ala-Cys-Gly-Arg	[C_14_H_27_N_7_O_5_S + K]^+^	444.14	444.14	0.0
**12.**	Phe-Trp-Gly	[C_22_H_24_N_4_O_4_ + K]^+^	447.14	447.14	0.0
**13.**	Asp-Gly-Arg	[C_12_H_22_N_6_O_6_ + ^107^Ag]^+^	453.07	453.06	22.1
**14.**	Ala-Ala-Ala-Asn	[C_13_H_23_N_5_O_6_ + ^109^Ag]^+^	454.07	454.07	0.0
**15.**	Dopaxanthin	[C_18_H_18_N_2_O_8_ + ^107^Ag]^+^	497.01	497.01	0.0
		[C_18_H_18_N_2_O_8_ + ^109^Ag]^+^	499.01	499.01	0.0
**16.**	CPA(18:2(9Z,12Z)/0:0)	[C_21_H_37_O_6_P + ^109^Ag]^+^	525.14	525.14	0.0
**17.**	CPA(18:1(11Z)/0:0)	[C_21_H_39_O_6_P + ^109^Ag]^+^	527.15	527.15	0.0
**18.**	Ile-Lys-Tyr	[C_21_H_34_N_4_O_5_ + ^107^Ag]^+^	529.16	529.16	0.0
**19**	PS(17:2(9Z,12Z)/0:0)	[C_23_H_42_NO_9_P + Na]^+^	530.25	530.25	0.0
**20.**	PA(17:0/0:0)	[C_20_H_41_O_7_P + ^107^Ag]^+^	531.16	531.16	0.0
**21.**	Arg-Ile-Ile-Pro	[C_23_H_43_N_7_O_5_ + K]^+^	536.29	536.30	−18.6
**22.**	PG(12:0/0:0)	[C_18_H_37_O_9_P + ^109^Ag]^+^	537.12	537.12	0.0
**23.**	LysoPC(18:4(6Z,9Z,12Z,15Z))	[C_26_H_46_NO_7_P + Na]^+^	538.29	538.29	0.0
**24.**	Ala-Glu-Ile-Thr	[C_18_H_32_N_4_O_8_ + ^107^Ag]^+^	539.13	539.13	0.0
**25.**	Asn-Leu-Phe-Phe	[C_28_H_37_N_5_O_6_ + H]^+^	540.28	540.28	0.0
**26.**	Ala-Phe-Val-Pro	[C_22_H_32_N_4_O_5_ + ^109^Ag]^+^	541.14	541.14	0.0
**27.**	His-Lys-Val-His	[C_23_H_37_N_9_O_5_ + Na]^+^	542.28	542.28	0.0
**28.**	Glu-Glu-Met-Pro	[C_20_H_32_N_4_O_9_S + K]^+^	543.15	543.15	0.0
**29.**	PC(O-10:1(9E)/2:0)	[C_20_H_40_NO_7_P + ^107^Ag]^+^	544.16	544.16	0.0
**30.**	LysoPA(0:0/18:1(9Z))	[C_21_H_41_O_7_P + ^109^Ag]^+^	545.16	545.16	0.0
**31.**	Asn-Trp-Asp-Pro	[C_24_H_30_N_6_O_8_ + Na]^+^	553.20	553.20	0.0
**32.**	Lys-Trp-Gly-Gly	[C_21_H_30_N_6_O_5_ + ^109^Ag]^+^	555.13	555.13	0.0
**33.**	Ala-Cys-Tyr-Tyr	[C_24_H_30_N_4_O_7_S + K]^+^	557.14	557.15	−17.9
**34.**	Arg-Asn-Asp-Gly	[C_16_H_28_N_8_O_8_ + ^107^Ag]^+^	567.11	567.11	0.0
**35.**	Phe-Thr-Pro-Pro	[C_23_H_32_N_4_O_6_ + ^109^Ag]^+^	569.14	569.14	0.0
**36.**	Thr-Trp-Arg	[C_21_H_31_N_7_O_5_ + ^109^Ag]^+^	570.14	570.14	0.0
**37.**	Ile-Met-Thr-Thr	[C_19_H_36_N_4_O_7_S + ^107^Ag]^+^	571.13	571.14	−17.5
**38.**	Cys-Ser-Tyr-Tyr	[C_24_H_30_N_4_O_8_S + K]^+^	573.14	573.14	0.0
**39.**	Phe-Gly-Ser-Tyr	[C_23_H_28_N_4_O_7_ + ^109^Ag]^+^	581.10	581.10	0.0
**40.**	Asn-Leu-Asn-Asp	[C_18_H_30_N_6_O_9_ + ^109^Ag]^+^	583.11	583.11	0.0
**41.**	Ala-Phe-Asn-Gln	[C_21_H_30_N_6_O_7_ + ^107^Ag]^+^	585.12	585.12	0.0
		[C_21_H_30_N_6_O_7_ + ^109^Ag]^+^	587.12	587.12	0.0
**42.**	His-Met-Phe-Gly	[C_22_H_30_N_6_O_5_S + ^107^Ag]^+^	597.10	597.10	0.0
**43.**	Arg-Cys-Gln-Ser	[C_17_H_32_N_8_O_7_S + ^107^Ag]^+^	599.12	599.12	0.0
		[C_17_H_32_N_8_O_7_S + ^109^Ag]^+^	601.12	601.12	0.0
**44.**	Cys-Met-Phe-Thr	[C_21_H_32_N_4_O_6_S_2_ + ^109^Ag]^+^	609.08	609.08	0.0
**45.**	Asp-Met-Thr-His	[C_19_H_30_N_6_O_8_S + ^109^Ag]^+^	611.09	611.09	0.0
**46.**	Ala-Trp-Cys-Gln	[C_22_H_30_N_6_O_6_S + ^107^Ag]^+^	613.10	613.10	0.0
		[C_22_H_30_N_6_O_6_S + ^109^Ag]^+^	615.10	615.10	0.0
**47.**	Asp-Met-Asp-His	[C_19_H_28_N_6_O_9_S + ^107^Ag]^+^	623.07	623.07	0.0
**48.**	Cys-Met-Thr-Tyr	[C_21_H_32_N_4_O_7_S_2_ + ^109^Ag]^+^	625.08	625.08	0.0
**49.**	Poly-g-D-glutamate	[C_20_H_30_N_4_O_12_ + ^109^Ag]^+^	627.09	627.09	0.0
**50.**	Asn-Met-Met-Gln	[C_19_H_34_N_6_O_7_S_2_ + ^107^Ag]^+^	629.10	629.10	0.0
**51.**	P-(bromoacetamido)phenyl uridylpyrophosphate	[C_17_H_20_BrN_3_O_13_P_2_ + Na]^+^	637.96	637.95	15.7
**52.**	Asn-Trp-Asp-Cys	[C_22_H_28_N_6_O_8_S + ^107^Ag]^+^	643.08	643.07	15.6
**53.**	TG(12:0/12:0/20:0)	[C_47_H_90_O_6_ + H]^+^	751.68	751.68	0.0
**54.**	Stigmast-5,22E-dien-3beta-yl (13Z-docosenoate)	[C_51_H_88_O_2_ + Na]^+^	755.67	755.67	0.0
		[C_51_H_88_O_2_ + K]^+^	771.64	771.64	0.0
**55.**	PE(15:0/22:0)	[C_42_H_84_NO_8_P + H]^+^	762.60	762.60	0.0
**56.**	GlcCer(d18:2/21:0)	[C_45_H_85_NO_8_ + H]^+^	768.64	768.63	13.0
**57.**	PS(17:1(9Z)/20:4(5Z,8Z,11Z,14Z))	[C_43_H_74_NO_10_P + K]^+^	834.47	834.47	0.0
**58.**	PG(18:1(9Z)/20:4(5Z,8Z,11Z,14Z))	[C_44_H_77_O_10_P + K]^+^	835.48	835.49	−12.0
**59.**	N-(2-hydroxy-eicosanoyl)-1-beta-glucosyl-4E,6E-pentadecasphingadienine	[C_41_H_77_NO_9_ + ^109^Ag]^+^	836.47	836.46	12.0
**60.**	PG(20:5(5Z,8Z,11Z,14Z,17Z)/20:5(5Z,8Z,11Z,14Z,17Z))	[C_46_H_71_O_10_P + Na]^+^	837.47	837.47	0.0
**61.**	TG(16:1(9Z)/18:3(6Z,9Z,12Z)/18:3(6Z,9Z,12Z))	[C_55_H_92_O_6_ + H]^+^	849.70	849.70	0.0
**62.**	[PC (3:0/3:0/3:0)] 1-(2E,6E,10E-phytatrienyl)-2-(2E,6E10E-phytatrienyl)-sn-glycero-3-phosphocholine	[C_48_H_88_NO_6_P + ^107^Ag]^+^	912.54	912.54	0.0
**63.**	PG(O-20:0/19:1(9Z))	[C_45_H_89_O_9_P + ^109^Ag]^+^	913.53	913.53	0.0
**64.**	PG(22:1(11Z)/22:6(4Z,7Z,10Z,13Z,16Z,19Z))	[C_50_H_85_O_10_P + K]^+^	915.56	915.55	10.9
**65.**	PC(22:4(7Z,10Z,13Z,16Z)/P-18:1(11Z))	[C_48_H_86_NO_7_P + ^107^Ag]^+^	926.51	926.52	−10.8
**66.**	PS(17:0/22:1(11Z))	[C_45_H_86_NO_10_P + ^109^Ag]^+^	940.51	940.50	10.6
**67.**	PIP(16:0/18:0)	[C_43_H_84_O_16_P_2_ + Na]^+^	941.51	941.51	0.0
**68.**	LacCer(d18:0/14:0)	[C_44_H_85_NO_13_ + ^107^Ag]^+^	942.51	942.51	0.0
**69.**	PI(17:2(9Z,12Z)/18:4(6Z,9Z,12Z,15Z))	[C_44_H_73_O_13_P + ^107^Ag]^+^	947.39	947.38	10.6
		[C_44_H_73_O_13_P + ^109^Ag]^+^	949.39	949.38	10.5
**70.**	PS(P-20:0/22:6(4Z,7Z,10Z,13Z,16Z,19Z))	[C_48_H_82_NO_9_P + ^107^Ag]^+^	954.49	954.48	10.5
		[C_48_H_82_NO_9_P + ^109^Ag]^+^	956.48	956.48	0.0
**71.**	PI(O-16:0/20:2(11Z,14Z))	[C_45_H_85_O_12_P + ^107^Ag]^+^	955.49	955.48	10.5
**72.**	PI(15:0/20:1(11Z))	[C_44_H_83_O_13_P + ^107^Ag]^+^	957.47	957.46	10.4
**73.**	PI(17:1(9Z)/20:3(8Z,11Z,14Z))	[C_46_H_81_O_13_P + ^109^Ag]^+^	981.44	981.45	−10.2
**74.**	TG(19:1(9Z)/21:0/21:0)	[C_64_H_122_O_6_ + H]^+^	987.94	987.93	10.1
**75.**	5-formyl-tetrahydrosarcinapterin	[C_36_H_52_N_7_O_20_P + ^107^Ag]^+^	1040.21	1040.21	0.0
**76.**	Lipid IVA	[C_68_H_126_N_2_O_23_P_2_ + H]^+^	1401.84	1401.83	7.1
**77.**	Tetrahexosylceramide (d18:1/26:1(17Z))	[C_70_H_128_N_2_O_23_ + K]^+^	1403.85	1403.85	0.0

* Putative metabolite.

**Table 6 foods-11-02257-t006:** Concentration of the analyzed vitamins in the niacin assay medium and the post-culture fluid of LAB strains obtained using HPLC analysis. During the analysis, the concentrations of the analyzed vitamins in the clean medium were taken into account (values for LAB strains shown in the table are after background subtraction). nd—not detected; LOD—limit of detection; LOQ—limit of quantification.

	Concentration (µg/mL)		
Sample	Niacin—B3	Folic Acid—B9	Riboflavin—B2
TEMPERATURE 20 °C			
Niacin Assay Medium	0.178 ± 0.009	nd	4.104 ± 0.126
*L. lactis*	3.582 ± 0.132	nd	1.409 ± 0.045
*W. cibaria*	nd	LOD	−2.276 ± 0.118
MRS Broth Medium	nd	nd	nd
*L. lactis*	0.718 ± 0.000	nd	0.147 ± 0.006
*W. cibaria*	1.465 ± 0.188	nd	0.109 ± 0.002
TEMPERATURE 30 °C			
Niacin Assay Medium	0.174 ± 0.001	nd	4.592 ± 0.095
*L. lactis*	1.273 ± 0.177	LOD	0.754 ± 0.065
*W. cibaria*	0.081 ± 0.005 LOQ	LOD	−0.307 ± 0.495
MRS Broth Medium	1.350 ± 0.073	nd	4.011 ± 0.067
*L. lactis*	−0.874 ± 0.072	nd	−3.871 ± 0.001
*W. cibaria*	−0.872 ± 0.058	nd	−1.314 ± 0.115

**Table 7 foods-11-02257-t007:** Concentration of the analyzed compounds in the LAPTg medium and the post-culture fluid of LAB strains obtained using HPLC analysis. During the analysis, the concentrations of the analyzed lactose and lactic acid in the clean medium were taken into account (values for LAB strains shown in the table are after background subtraction).

Sample	Lactose (µg/mL)	Lactic Acid (µg/mL)
TEMPERATURE 20 °C		
LAPTg Medium	38.61 ± 0.58	34.01 ± 1.88
*L. lactis*	2.61 ± 0.81	1014.99 ± 47.59
*W. cibaria*	14.49 ± 1.14	310.99 ± 7.48
TEMPERATURE 30 °C		
LAPTg Medium	49.91 ± 1.00	16.43 ± 0.22
*L. lactis*	12.38 ± 0.95	638.21 ± 8.03
*W. cibaria*	0.91 ± 0.50	277.36 ± 5.82

**Table 8 foods-11-02257-t008:** Results of the experiment aimed at examining the antagonistic properties of LAB strains against pathogenic bacteria.

Pathogenic Strain	Number of Grown Pathogen Colonies (Percentage of Inhibition)		
	Control	*L. lactis*	*W. cibaria*
*Escherichia coli*	340 ± 63	174	156
		49%	54%
*Enterobacter cloacae*	62 ± 13	22	5
		65%	92%
*Pseudomonas aeruginosa*	204 ± 12	0	0
		100%	100%
*Listeria innocua*	956 ± 35	0	0
		100%	100%

## Data Availability

All sequence data that support the findings of this study have been deposited in GenBank and the public URL for the sequence collection is https://www.ncbi.nlm.nih.gov/sites/myncbi/1F1i8raBplj5d/collections/61883288/public/ (accessed on 1 June 2022) *Lactococcus lactis* strain Beet1 (OM957496) and *Weissella cibaria* strain Beet2 (OM957500).

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
