# Peer review of "Isolation and Identification of Lactococcus lactis and Weissella cibaria Strains from Fermented Beetroot and an Investigation of Their Properties as Potential Starter Cultures and Probiotics"

_foods, 2022, doi:10.3390/foods11152257_

Round 1
Reviewer 1 Report
This paper describes the evaluation of lactic acid bacteria isolated from fermented vegetable products. Overall, the article presents an extensive work of analysis aiming to determine if the selected strains have biotechnological potential for different applications. In that sense, the study is a good effort to assess new LAB strains. However, various aspects of the study and writing must be improved or clarified:
-The first aspect deals with the aim of the study. The paper describes the assessment of probiotic properties of selected isolates. However, the experiments are directed towards the evaluation of characteristics that are mostly important for starter cultures (that are not necessarily good probiotics). Probiotic tests may include resistance to pH, resistance to bile salts, adhesion properties, cell culture assays, safety assessment, etc. This aspect must be clarified throughout the text.
-Pickled beet root was used as the source to isolate the LAB strains for the study. I think it would help to briefly describe the product that you used. Is this a product common in your country? Is there something important about pickled beet root that made you think it was a good source to isolate new strains with biotechnological potential? What kind of sample did you use? Was it commercial or homemade?
-A key aspect is that I did not understand what the criteria was to select L. lactis and W. cibaria. In line 121 you mentioned, "2 with the expected properties" were selected. But what are the expected properties? Why other isolates were discarded?
-Section 2.5 described "Determination of the optimal conditions for growth". I am not sure that you present an optimization experiment. Usually, this is done through a different experimental approach and design such as Central Composite Design. I suggest renaming this section as just "Growth assessment of selected isolates". In this same experiment, no statistics were used or at least it was not mentioned like that. Still, some conclusions were made regarding which media or temperature was better or worst. In line 311, it is mentioned that "For the analysis of the growth rate...."; I do not see any growth rate calculation and I do not think you have complete data to estimate that. Discussion of this section is incomplete as it lacks of context. No comparison with any previous study was made.
-Section 2.8 describes the analysis of fats. This part is interesting as you mentioned that normally this kind of assessment is not performed on new LAB strains. However, I still did not understand what was the purpose of that considering the objective of research. It seems it was a lot of analytical effort dedicated into a section that (in my opinion) did not contribute greatly to the discussion. More elements are necessary to convince other researchers to pursue this type of assessment on new strains.
-Section 2.10 presents the experiments to determine antimicrobial activity of LAB supernatant. I strongly recommend to adjust these results and improve the experiment. Here are some recommendations:
a.The best way to assess for the presence of antimicrobial compounds in LAB supernatants is by neutralizing acidic pH normally associated with this type of sample. Lactic acid and others will be always there and of course you will observe inhibition. By neutralizing pH, the results will demonstrate the presence of pure antimicrobial compounds. Do not forget to filter the supernatant before the experiments (not mentioned in the methodology part).
b. I do not see the need for three different ways of evaluating antimicrobial activity on solid media. You are basically observing the same behavior and phenomena. In addition, the first approach seems too qualitative and I do not think it provides enough background for conclusions. I would rather eliminate those results from the manuscript and I guess just one of the 3 experiments is enough (I think the use of paper discs). What was the control in this experiment?
c. A second experiment could be the evaluation of the supernatant effect on liquid media. There are several references that show the methodology to perform this kind of experiments. The classic way is by using a 96-well microplate to expose each pathogen to a certain volume (or several) of the supernatant.
-In line 248 you mentioned that you used 5 pathogenic bacteria but Listeria innocua is not a human pathogen.
-Table 8 shows the inhibition zones for experiments 2 and 3 and a classification is mentioned according with each result. However, it is not clear where this criteria comes from.
-From lines 634 to 648 you present a sinopsis of what was done in this research. However, there are no conclusions at all. It must be changed to fit a "Conclusions" section.
-The novelty of the study is not completely clear. This type of research is very similar to other publications already in literature. I suggest to incorporate additional elements to convince the public about the uniqueness of your work.
-In general, the writing must be improved. Considering the number of authors, it is likely that different persons took part on the writing process and different styles can be perceived just by reading the paper. In some parts English is better but there are important deficiencies in other sections of the paper body. There is lack of writing uniformity and a professional in English could help to compensate for this. There are several typos and errors but the most common was skipping the use of italics for scientific names; this happens many times throughout the entire body text. Also, try to avoid the use of "first person".
Author Response
Answer to the Reviewers’ Comments and Changes Made
We would like to thank the Reviewers for careful reading, and constructive suggestions for our manuscript that will help us to improve our work. According to the comments from the reviewers, we comprehensively revised our manuscript. Hoping that we addressed all the questions mentioned by the reviewers, below we include the point-to-point response to each comment.
In the manuscript file (MARKED UP MANUSCRIPT) all the changes have been provided by using the track change mode in Word.
Reviewer #1
Comment: This paper describes the evaluation of lactic acid bacteria isolated from fermented vegetable products. Overall, the article presents an extensive work of analysis aiming to determine if the selected strains have biotechnological potential for different applications. In that sense, the study is a good effort to assess new LAB strains. However, various aspects of the study and writing must be improved or clarified:
-The first aspect deals with the aim of the study. The paper describes the assessment of probiotic properties of selected isolates. However, the experiments are directed towards the evaluation of characteristics that are mostly important for starter cultures (that are not necessarily good probiotics). Probiotic tests may include resistance to pH, resistance to bile salts, adhesion properties, cell culture assays, safety assessment, etc. This aspect must be clarified throughout the text.
Answer: Thanks the Reviewer for a comment. Indeed, when we write probiotic properties, we must emphasize that we mean the potential probiotic properties of the selected strains based on scientific reports on the species studied. Perhaps the reviewer was right that in the context of our article, we should clarify that currently their properties as starter cultures for future use in functional foods production were mainly studied. We have therefore revised the entire paper to clarify this point.
Comment: -Pickled beet root was used as the source to isolate the LAB strains for the study. I think it would help to briefly describe the product that you used. Is this a product common in your country? Is there something important about pickled beet root that made you think it was a good source to isolate new strains with biotechnological potential? What kind of sample did you use? Was it commercial or homemade?
Answer: We thank the Reviewer for a comment. In our research, homemade pickled beet root was used to find new wild LAB strains that will show beneficial properties as future starter cultures with probiotic potential. Such foods are widely regarded in Poland as having one of the most health-promoting properties and a good source of natural probiotics. Therefore, we thought it worthwhile to use pickled beet root as a source for the isolation and selection of LAB strains. The description of the LAB source selection was added to the Materials and Methods section.
Comment: -A key aspect is that I did not understand what the criteria was to select L. lactis and W. cibaria. In line 121 you mentioned, "2 with the expected properties" were selected. But what are the expected properties? Why other isolates were discarded?
Answer: We thank the Reviewer for his insightful comments. When selecting the strains, we kept in mind their potential future use. L. lactis is a well-known LAB strain that shows high potential as a probiotic, as well as its use in food production has a long history. Nevertheless, L. lactis is mainly isolated from dairy products. In our work, we focused on another type of potential probiotic source, namely silage. We assumed that strains obtained from this source may exhibit new and desirable characteristics. The second strain, W. cibaria, was found to have probiotic potential and various beneficial properties, including functional production of exopolysaccharides and attenuation of pathogen-induced inflammatory responses. In addition, this strain has been found to be the dominant species in the microflora of silage products such as kimchi [see Yu et al. J. Microbiol. Biotechnol. (2019),29(7), 1022–1032 https://doi.org/10.4014/jmb.1903.03014]. Therefore, we surmised that it would be beneficial to select this strain for further analysis, especially since we are also isolating the bacteria from silages. Suitable comment about criteria for strains selection was added to the manuscript.
Comment: Section 2.5 described "Determination of the optimal conditions for growth". I am not sure that you present an optimization experiment. Usually, this is done through a different experimental approach and design such as Central Composite Design. I suggest renaming this section as just "Growth assessment of selected isolates". In this same experiment, no statistics were used or at least it was not mentioned like that. Still, some conclusions were made regarding which media or temperature was better or worst. In line 311, it is mentioned that "For the analysis of the growth rate...."; I do not see any growth rate calculation and I do not think you have complete data to estimate that. Discussion of this section is incomplete as it lacks of context. No comparison with any previous study was made.
Answer: We thank the Reviewer for insightful comments. Indeed, the growth analysis performed in the submitted manuscript does not fit sensu stricto into the definition of an optimization experiment. Therefore, we agree that "Growth assessment of selected isolates" will be more suitable title of this part of the experiment. The aim of this step was to find the best culture conditions for the selected bacteria cultivation in view of the biomass preparation for future starter cultures. Regarding this, obtained results were enough to set the best conditions for obtaining the highest biomass yield and there was no point to perform additional calculation. The purpose of this stage of the study was not to compare with other strains and works but only to find the best conditions, which was achieved.
Comment: -Section 2.8 describes the analysis of fats. This part is interesting as you mentioned that normally this kind of assessment is not performed on new LAB strains. However, I still did not understand what was the purpose of that considering the objective of research. It seems it was a lot of analytical effort dedicated into a section that (in my opinion) did not contribute greatly to the discussion. More elements are necessary to convince other researchers to pursue this type of assessment on new strains.
Answer: We thank the Reviewer for insightful comments. Lipidomic analysis was carried out for lactic acid bacteria, because LABs are an important source of lipids. Lipid compounds play a key role in the stabilization of cell membranes (they are a building block of membranes), they take part in the transport of proteins, DNA replication, and are a reserve and energy material for the cell. Due to a number of different properties of lipids, a lipidomic analysis was performed in order to determine the lipid profile for the tested systems. As a result of the research, it was possible to determine the lipid composition and changes in the lipid composition were observed for the tested bacterial strains. Examination of the lipid profile of bacteria is very important because any change in the lipid composition affects, among others, on the activity of cytoplasmic proteins and contributes to the adaptation of bacteria to the environment.
Comment: -Section 2.10 presents the experiments to determine antimicrobial activity of LAB supernatant. I strongly recommend to adjust these results and improve the experiment. Here are some recommendations:
a.The best way to assess for the presence of antimicrobial compounds in LAB supernatants is by neutralizing acidic pH normally associated with this type of sample. Lactic acid and others will be always there and of course you will observe inhibition. By neutralizing pH, the results will demonstrate the presence of pure antimicrobial compounds. Do not forget to filter the supernatant before the experiments (not mentioned in the methodology part).
- I do not see the need for three different ways of evaluating antimicrobial activity on solid media. You are basically observing the same behavior and phenomena. In addition, the first approach seems too qualitative and I do not think it provides enough background for conclusions. I would rather eliminate those results from the manuscript and I guess just one of the 3 experiments is enough (I think the use of paper discs). What was the control in this experiment?
- A second experiment could be the evaluation of the supernatant effect on liquid media. There are several references that show the methodology to perform this kind of experiments. The classic way is by using a 96-well microplate to expose each pathogen to a certain volume (or several) of the supernatant.
Answer: Thanks to the Reviewer for the valuable suggestions. Indeed, pH will demonstrate inhibition effect on the growth of the pathogenic strains growth. As we wrote in the manuscript produced lactic acid will contribute to the inhibition of the growth of pathogenic and food spoilage microorganisms, similar to other organic acids produced by LAB by lowering the pH of the environment, neutralizes the electrochemical potential of cell membranes and denaturates intracellular proteins of microorganisms. Antimicrobial activity of probiotic strains also results from their ability of forming biofilms on the gut walls, competing for nutrients, and by producing or releasing various metabolites (beside lactic acid, also antimicrobial peptides, short chain fatty acids, and hydrogen peroxide). We agree with the Reviewer comment, that obtained results in our studies do not provide information about mechanism of antimicrobial action of the selected LAB strains, however, at this stage of the studies the main goal is the selection of the strains with the highest potential as future starter cultures with probiotic potential. Therefore, our intention was not to describe the mechanism of growth inhibition of pathogenic flora under the influence of selected strains, but rather to assess whether the inhibitory effect occurs or not. Since, as we wrote, the production of the organic acid can inhibit growth of the bacteria not only by decreasing pH but also by interfering with microbial proteins and changing electrochemical potential of cell membranes, we decide to measure the global inhibitory effect. Nevertheless, the indication of the presence of antimicrobial compounds, such as bacteriocins, is very important and will be studied in detail in the next stages of the research, when the probiotic potential of the selected strains will be examined sensu stricto. Regarding other reviewer's suggestions: we agree that only results from one experiment could be presented - we decide to leave the results from the first experiment; the experiment using a 96-well microplate and liquid media is interesting and will be used in the next step when the strict mechanisms of antimicrobial action will be studied.
Comment: -In line 248 you mentioned that you used 5 pathogenic bacteria but Listeria innocua is not a human pathogen.
Answer: Thanks the Reviewer for the insightful comment. Listeria innocua was used as food spoilage microorganism – we missed this information. The sentence was revised to clarify this issue.
Comment: -Table 8 shows the inhibition zones for experiments 2 and 3 and a classification is mentioned according with each result. However, it is not clear where this criteria comes from.
Answer: Taking into account an earlier suggestion, the results from these two experiments were removed due to their repeating the information from the first experiment.
Comment: -From lines 634 to 648 you present a sinopsis of what was done in this research. However, there are no conclusions at all. It must be changed to fit a "Conclusions" section.
Answer: Thanks the Reviewer for the insightful comment. We have corrected Conclusions according to comments.
Comment: -The novelty of the study is not completely clear. This type of research is very similar to other publications already in literature. I suggest to incorporate additional elements to convince the public about the uniqueness of your work.
Answer: As we underlined now in section Conclusions we have isolated new strains with great biotechnological potential considering many aspects i.a. proteolytic activity, synthesis of vitamins B and lactic acid, inhibition of pathogens growth. We also proposed organic volatile compounds analysis which allowed as to estimate organoleptic properties of potential dairy products. One of the most interesting studies that will be continued during later studies is lipidomic analysis. As we mentioned earlier in response LABs are an important source of lipids. Lipid compounds play a key role in the stabilization of cell membranes (they are a building block of membranes), they take part in the transport of proteins, DNA replication, and are a reserve and energy material for the cell. Due to a number of different properties of lipids, a lipidomic analysis was performed in order to determine the lipid profile for the tested systems. As a result of the research, it was possible to determine the lipid composition and changes in the lipid composition were observed for the tested bacterial strains. Examination of the lipid profile of bacteria is very important because any change in the lipid composition affects, among others, on the activity of cytoplasmic proteins and contributes to the adaptation of bacteria to the environment.
In our previous studies (ZÅ‚och et al., 2022) we showed that LAB can acts as modulator of fatty acids composition in dairy raw material (for example cream) and hence can influence on fatty acid composition of the product (butter). The addition of this lactic acid bacteria (Lacticaseibacillus paracasei) enriched the cream in 9-hexadecenoic acid, oleic acid, octadeca-9,12-dienoic acid, or conjugated linoleic acid that exhibit antimutagenic and anticarcinogenic properties. Moreover, higher level of monounsaturated fatty acids can extend the shelf life of the butter in the future. This topic is very interesting for us and presented results (isolation, identification and characterization of L. lactis and W. cibaria) will be extended to research on the possibility of using the isolated strains to modulate the fatty acid profile.
Złoch, M.; Rafińska, K.; Sugajski, M.; Buszewska-Forajta, M.; Walczak-Skierska, J.; Railean, V.; Pomastowski, P.; Białczak, D.; Buszewski, B. Lacticaseibacillus paracasei as a Modulator of Fatty Acid Compositions and Vitamin D3 in Cream. Foods 2022, 11, 1659. https://doi.org/10.3390/foods11111659
We would like to also underline that Reviewer#2 indicates that some of the techniques such as MALDI and SALDI looked excellent, innovative, and new.
Comment: -In general, the writing must be improved. Considering the number of authors, it is likely that different persons took part on the writing process and different styles can be perceived just by reading the paper. In some parts English is better but there are important deficiencies in other sections of the paper body. There is lack of writing uniformity and a professional in English could help to compensate for this. There are several typos and errors but the most common was skipping the use of italics for scientific names; this happens many times throughout the entire body text. Also, try to avoid the use of "first person".
Answer: Thanks the Reviewer for the suggestion. English and style were improved (as much as we could).

Reviewer 2 Report
The workload was high and some of the techniques such as MALDI and SALDI looked excellent, innovative, and new.
It is better to state in the title that these two bacterial strains were isolated during this research work. e.g. " Isolation and identification of Lactococcus lactis and Weissella cibaria strains from fermented beetroot and study on ............"
Line 94 states that these isolates could be used in the production of dairy products, so it would be better to study the production of vitamins and aromatic compounds in a milk-based medium.
Author Response
Answer to the Reviewers’ Comments and Changes Made
We would like to thank the Reviewers for careful reading, and constructive suggestions for our manuscript that will help us to improve our work. According to the comments from the reviewers, we comprehensively revised our manuscript. Hoping that we addressed all the questions mentioned by the reviewers, below we include the point-to-point response to each comment.
In the manuscript file (MARKED UP MANUSCRIPT) all the changes have been provided by using the track change mode in Word.
Reviewer #2
Comment: The workload was high and some of the techniques such as MALDI and SALDI looked excellent, innovative, and new.
Answer: We thank the Reviewer for appreciating our idea and work. Indeed, the use of MALDI and SALDI in microbiological research and selection of beneficial LAB strains is both a great novelty and opens up the possibility of furthering our knowledge of potential probiotics and starter culture bacteria.
Comment: It is better to state in the title that these two bacterial strains were isolated during this research work. e.g. " Isolation and identification of Lactococcus lactis and Weissella cibaria strains from fermented beetroot and study on ............"
Answer: Based on the Reviewer suggestion the title of the work has been changed into: “Isolation and identification of Lactococcus lactis and Weissella cibaria strains from fermented beetroot and study on their properties as potential starter culture and probiotics”.
Comment: Line 94 states that these isolates could be used in the production of dairy products, so it would be better to study the production of vitamins and aromatic compounds in a milk-based medium.
Answer: Thanks the Reviewer for the suggestion. The presented work is focused on the selection of the strains obtain from a relatively rare source of the LAB strains for future application in the dairy industry. For this, firstly properties of the selected strains were examined in the in vitro conditions using common microbial buffers and media. Now that the potential of the tested strains has been assessed, the next step will be to see if the observed potential will be achievable in the dairy products what will be the issue of the next studies.

Round 2
Reviewer 1 Report
I do not hace additional comments for the authors. The article was improved and all the issues were addressed or clarified.